# Social structure learning in human anterior insula

Tatiana Lau[1], Samuel J Gershman[2], Mina Cikara[2]*

[1]Royal Holloway, University of London, Egham, United Kingdom; [2]Harvard University, Cambridge, United States

**Abstract** Humans form social coalitions in every society, yet we know little about how we learn and represent social group boundaries. Here we derive predictions from a computational model of latent structure learning to move beyond explicit category labels and interpersonal, or *dyadic*, similarity as the sole inputs to social group representations. Using a model-based analysis of functional neuroimaging data, we find that separate areas correlate with dyadic similarity and latent structure learning. Trial-by-trial estimates of 'allyship' based on dyadic similarity between participants and each agent recruited medial prefrontal cortex/pregenual anterior cingulate (pgACC). Latent social group structure-based allyship estimates, in contrast, recruited right anterior insula (rAI). Variability in the brain signal from rAI improved prediction of variability in ally-choice behavior, whereas variability from the pgACC did not. These results provide novel insights into the psychological and neural mechanisms by which people learn to distinguish 'us' from 'them.'

## Introduction

Being able to distinguish 'us' from 'them' is a core social capacity (*Brown, 1991*). In an increasingly interconnected world where people have multiple intersecting identities that guide their thoughts, feelings, and behaviors, being able to differentiate friend from foe is of paramount importance. Yet we know surprisingly little about how social group boundaries are learned and represented in the brain—particularly in the absence of overt cues to individuals' group membership.

One dominant account is that people use judgments of similarity to one's self on some contextually relevant feature (e.g., skin tone). Accordingly, neuroimaging studies have attempted to identify an overlap between brain regions associated with self-referential processes and categorization of others as in-group members (*Molenberghs and Morrison, 2014*; *Morrison et al., 2012*) A ventral region of medial prefrontal cortex (vmPFC), including pregenual anterior cingulate cortex (pgACC), is reliably associated with thinking about one's own and similar others' traits, mental states, and characteristics (*Cikara et al., 2014*; *Heleven and Van Overwalle, 2016*; *Jenkins et al., 2008*). But are similarity-based estimates sufficient for categorizing others as in-group versus out-group members and informing subsequent behavior?

Classic social psychological theories of intergroup relations indicate that there are other dimensions by which groups are defined (*Sherif, 1966*). Rather than prioritizing similarity to oneself, people may rely on functional relations between one's self and a target (e.g., 'Are you with me or against me?'; *Cikara and Fiske, 2013*).

Given that social categorization is such a flexible, dynamic process, how do people accumulate group structure information (especially in the absence of overt cues to group membership)? On one hand, they might try to characterize their ties with each individual (e.g., how well do I get along with Sue, with Dan, etc.). However, social group dynamics may be better captured by a model that integrates information about how agents relate to one another in addition to oneself (e.g., how do Sue and Dan get along with each other, and how do I get along with either of them?), which would allow perceivers to infer social latent group structure.

*For correspondence:
mcikara@fas.harvard.edu

Competing interests: The authors declare that no competing interests exist.

**eLife digest** In every society, people form social coalitions — we draw boundaries between 'us' and 'them'. But how do we decide who is one of 'us' and who is one of 'them'? One way is to use arbitrary categories. For example, we say that those living 49 degrees north of the Earth's equator are Canadian, whereas those living south of it are American. Another possibility is to use physical characteristics. But what about when neither of these options are available?

By monitoring brain activity in healthy volunteers learning about other people's political values, Lau et al. obtained insights into how people make these decisions. Participants lying in a brain scanner were asked to report their position on a political issue. They then learned the positions of three other hypothetical participants – A, B and C – on the same issue. After repeating this procedure for eight different issues, the volunteers had to decide whether they would align with A or with B on a 'mystery' political issue.

So how do participants choose between A and B? One possibility is that they simply choose whichever one has views most similar to their own. If this is the case, the views of hypothetical person C should not affect their decision. But in practice, C's views – specifically how much they resemble the volunteer's own – do influence whether the volunteer chooses A or B. This suggests that we choose our allies based on more than just their similarity to ourselves.

Using a mathematical model, Lau et al. show that volunteers also take into account how similar the views of the other 'participants' are to each other. In other words, they consider the structure of the social group as a whole. Moreover, the results from brain imaging show that different regions of the brain are active when volunteers track the structure of the entire group, as opposed to their own similarity with each individual.

Notably though, the activity of the group-tracking region explains people's alignment choices better than the activity of the similarity-tracking region. This suggests that we base our judgments of 'us' versus 'them' more on the structure of the group as a whole than on our own similarity with individual group members. Understanding how we determine whether others are on the same 'team' as ourselves could ultimately help us find ways to reduce bias and discrimination between groups.

---

If people represent social latent group structure (*Figure 1A*) in addition to dyadic similarities, then even when two agents' choices (Agents A's and B's) are equally similar to their own, the presence of a third agent (Agent C) altering the group structure should influence their decisions (*Figure 1B*). Importantly, dyadic similarity accounts would not predict differential ally-choice behavior in these cases (because similarity is equated for the two agents in question). In other words, the key difference between the two models is whether or not the presence of the third agent can affect how the first two agents are perceived.

To determine whether people possess different neural mechanisms for dyadic similarity and latent group structure learning, we created a structure-learning task in which participants reported their own position on a political issue and then guessed and learned via feedback the positions of three other agents, Agents A, B, and C, on the same issue (*Figure 2A*). After repeating this for eight political issues, participants were shown pictures of two of the three agents and asked to indicate with which of the two agents, Agent A or Agent B, they would align on an unknown political issue (*Figure 2B*; *Lau et al., 2018*). We focused on political issues because recent evidence suggests that implicit bias and behavioral discrimination along political party lines is now as potent as bias against racial out-groups (*Iyengar et al., 2012*; *Iyengar and Westwood, 2015*).

This design allowed us to i) investigate participants' trial-by-trial alignment signals based on dyadic similarity (with each respective agent), feature similarity-over-agents, and social latent structures, and ii) identify which brain regions tracked each of these representations. We then tested whether variability in the neural signal associated with these representations improved prediction of variability in participants' ally-choice behavior.

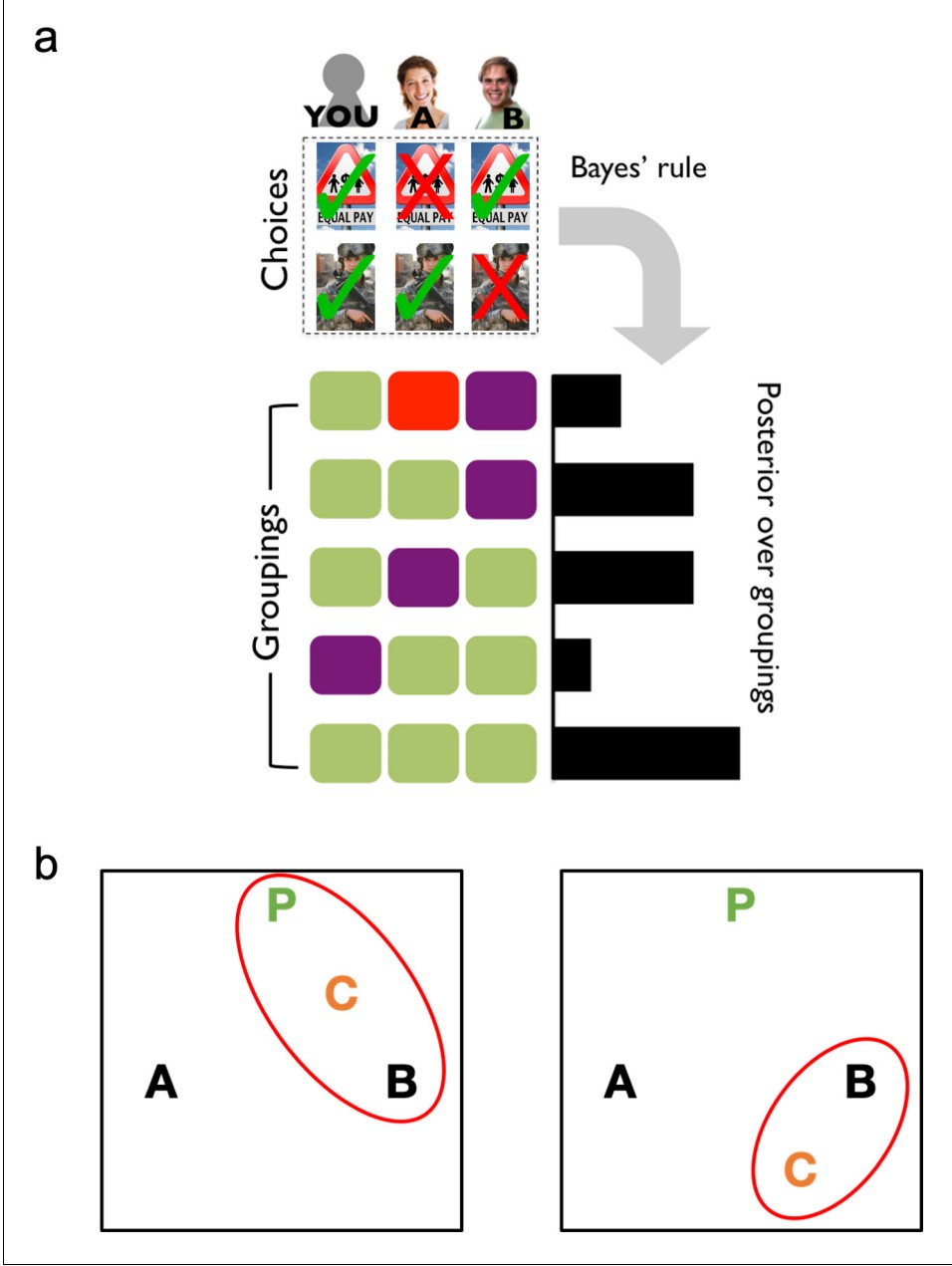

**Figure 1.** A formal account of social latent structure learning. (**A**) Model schematic illustrating how choice patterns are transformed using Bayes' rule to create a posterior over different possible latent groupings of agents. (**B**) Agents are represented as letters in an abstract space (P is the participant), where the distance between letters indicates the degree to which agents agree in their choices (i.e., choice overlap). Red ovals indicate the latent structures that have high posterior probability. Left: The placement of Agent C creates a cluster that includes both the participant and Agent B, which should increase estimates of Agent B as an ally. Right: The placement of Agent C excludes the participant from the cluster with Agents B and C, which should decrease estimates of Agent B as an ally.

## Results

We scanned 42 participants using functional magnetic resonance imaging (fMRI) as they completed our structure-learning task. Each participant completed six runs; each run comprised learning about three novel agents across eight political issues and then choosing to ally with one of two agents on

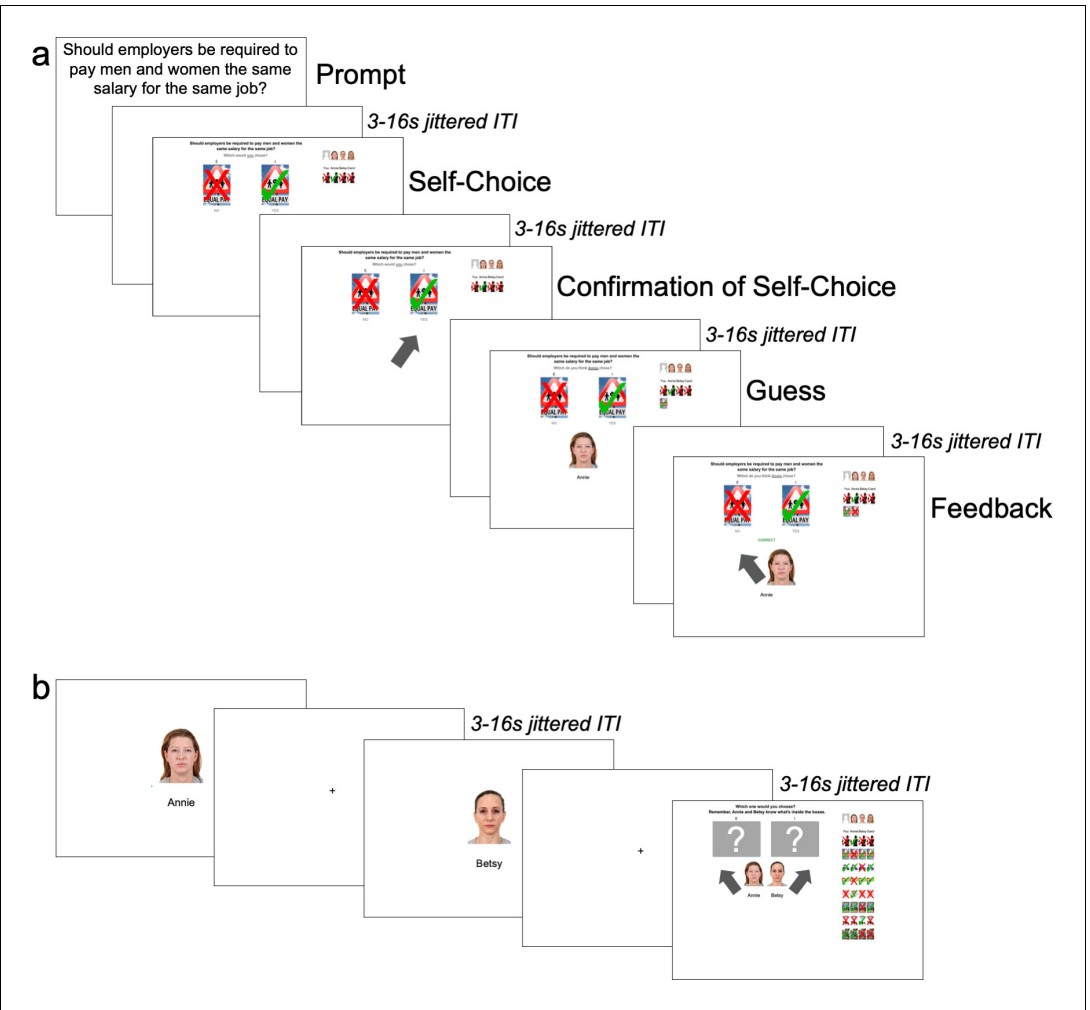

**Figure 2.** Order of events in task. (**A**) Learning Trials: Participants began every trial by seeing a political issue and reporting their personal stance on it. After receiving confirmation of their choice, they then guessed and received feedback on how the first agent responded to the same political issue and repeated this for the other two agents before moving onto a new political issue. (**B**) Ally-choice Trial: After eight learning trials, participants saw photos of Agents A and B sequentially in a random order and chose to align with either Agent A or B on a 'mystery' (i.e., unknown) political issue.

an unknown, 'mystery' issue. Each participant saw 48 political issues and learned about 18 novel agents in total.

## Learning about other agents' policy preferences and ally-choice trials

To model the probability of choosing Agent B's choice in the ally-choice trial as a function of social latent structure, we used a logistic regression predicting whether our participants in the scanner (N = 42) chose Agent B's choice during the ally-choice trial as a function of Agent B's agreement and Agent C's agreement with the participant. (See Materials and methods for analysis of the full N = 333 sample.) Because Agent A's preferences were always the inverse of Agent B's, including Agent A's agreement would have created a multicollinearity problem (recall also that participants could only choose either Agent B or Agent A). Including random slopes to account for subject-level effects resulted in a singular fit of the model (i.e., overfitting), so we removed them. We compared the full model including both main effects and the interaction with simpler models (including only main effects or including only Agent B's agreement with the participant). Likelihood-ratio tests indicated that the fully saturated model with both main effects and the interaction term fit the data better than without the interaction term ($\chi^2(1) = 7.246$, p=0.007).

Replicating previous behavioral results (*Lau et al., 2018*), we found that increasing Agent C's alignment with the participant made respondents more likely to choose Agent B on the ally-choice trial, above and beyond the participant's similarity with Agents A and B (*Figure 3*). As a simple dyadic similarity account would predict, the model indicated a significant positive effect of Agent B's agreement in predicting the likelihood of choosing Agent B in the ally-choice trial, $b = 2.325$, Wald's $z = 4.099$, 95% CI [1.284, 3.519], p<0.001. However, as predicted by the latent structure learning account, the model also indicated a significant positive effect of Agent C's agreement in predicting the likelihood of choosing Agent B in the ally-choice trial, $b = 1.322$, Wald's $z = 2.633$, 95% CI [0.371, 2.349], p=0.008 (*Figure 3*). This was qualified by a significant negative interaction between the agreements of Agent B and Agent C, $b = -0.307$, Wald's $z = -2.588$, 95% CI [−0.550,–0.082], p=0.010.

In other words, even when adjusting for Agent B's agreement with the participant, increasing Agent C's alignment with the participant made respondents more likely to choose Agent B on the ally-choice trial. However, this result was expectably qualified by a weak interaction: when Agent B agreed with the participant a majority of the time, the additional variance explained by Agent C's agreement decreased. While a dyadic similarity model would not predict that the level of agreement

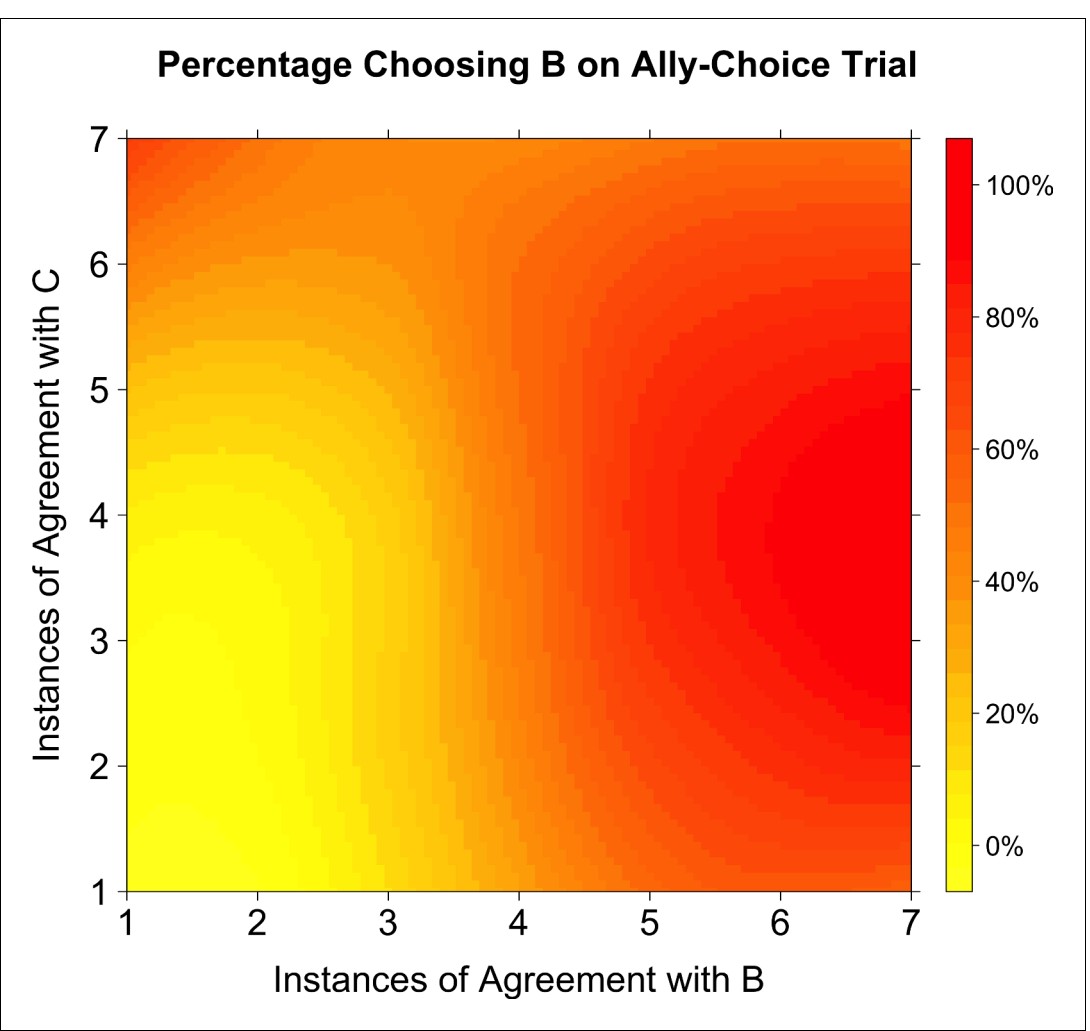

**Figure 3.** Percentage choosing Agent B as a function of agreement with Agents B and C. A smoothed level plot illustrates that as agreement with both Agents B and C increases (towards the top-right corner), so does the probability of choosing Agent B on the ally-choice trial. If Agent B had been the only influence on whether or not participants chose Agent B on the ally-choice trial, then the transition from yellow to red should only occur in the horizontal direction (as agreement with Agent B increases). Instead, there is a radial transition from the bottom-left corner.

with Agent C should matter for choosing on the ally-choice trial, the latent structure learning model does predict that Agent C's level of agreement with the participant should matter in whether or not participants choose Agent B on the ally-choice trial. Indeed, any difference in choice behavior as a result of Agent C's level of agreement is already inconsistent with the dyadic similarity account. Disambiguation between these two accounts of ally-choice have been demonstrated previously with model simulations and behavioral studies (*Gershman et al., 2017*; *Lau et al., 2018*).

## Computational models and neuroimaging data

We developed three models to capture participants' trial-by-trial estimates of i) dyadic similarity with each agent, ii) similarity-over-agents, and iii) social latent structure.

We calculated dyadic similarity $S_d$ as a function of the number of previous agreement instances between the agent under consideration and the participant divided by the number of trials elapsed, initialized at 0.50 for each agent (see Materials and methods for model details). For example, if an agent agreed with the participant on the first political issue, $S_d$ would be calculated as 0.66 for the second issue; if the agent did not agree with the participant on the first political issue, $S_d$ would be calculated as 0.33 for the second issue.

Unlike the latent structure learning model described below, the dyadic similarity model did not account for the feedback of other agents when calculating $S_d$. In other words, for the third trial, the dyadic similarity model would calculate a similarity for an agent who had agreed twice with the participant on the previous two trials as 0.75, regardless of how the other two agents had responded. On each new run, $S_d$ for each agent was set to 0.50 given that participants had no information about those three new agents. *As such, this value reflected how likely each agent was to agree with the participant on a new issue given that agent's agreement with the participant on previous issues.*

The person-as-feature similarity model calculated $S_f$ as the correlation between rows of a similarity matrix constructed from all possible dyadic similarities (i.e., $S_d$ between the participant and each agent as well as $S_d$ between each pair of agents). In other words, $S_f$ could be considered a second-order dyadic similarity in that it captures similarity over people rather than over choices. *As such, this value reflected how likely each agent was to agree with the participant given that the agent's and the participant's political preferences similarly resembled those of other agents.*

In contrast, the latent structure learning model (*Gershman et al., 2017*) assumes that participants infer latent group assignments (a partition of agents into groups) on the basis of the agents' choice data (see Materials and methods for model details). This model uses the Chinese restaurant process (*Aldous, 1985*) as a prior over group assignments, which effectively 'infers' the most probable number of clusters in the environment given the existing data, therefore bypassing the need to *a priori* set an expected number of clusters (e.g., one rarely walks into a room expecting there to be *n* number of groups). Through the observations of agents' choices, the posterior is inferred using Bayes' rule, and the likelihood, derived from analytically marginalizing the latent parameters under a Dirichlet-Multinomial model, will favor groupings where individuals in the same group exhibit similar choice patterns. Parametric modulator values for the latent structure learning model were calculated as the marginal posterior probabilities of relevant partitions (i.e., partitions wherein the participant and the respective agent were grouped together). To generate values, we did not fit any free parameters to participant behavior. Unlike our other two models, information about all three agents contributed on each trial to the prior for guessing about each particular agent. In other words, the prior for the second agent during the third trial took into account the feedback for the first agent in the third trial as well as the feedback for all three agents during the first and second trials. *As such, this value reflected how likely each agent was to belong to the same social group as the participant based on how all of the agents and the participant related to one another on previous issues.*

We examined which voxels' signal correlated with the dyadic similarity, feature similarity-over-agents, and latent structure learning parametric modulator contrasts (*Table 1*; no other regions other than the ones reported here exceeded our corrected threshold.). As predicted, trial-by-trial dyadic similarity correlated with activity in the ventral medial prefrontal cortex/pregenual anterior cingulate (pgACC; *Figure 4*, green). The similarity-over-agents modulator identified clusters in the pgACC, bilateral temporoparietal junction, right superior temporal sulcus, and left supplementary motor area (*Figure 4*, yellow). Perhaps unsurprisingly, the pgACC cluster from this parametric modulator encompassed the pgACC cluster found by the dyadic similarity modulator.

**Table 1.** Results from parametric modulator contrasts.

| Model | Region | Cluster size | X | Y | Z |
|---|---|---|---|---|---|
| *Dyadic Similarity* | Pregenual Anterior Cingulate | 327 | 18 | 48 | 0 |
| *Feature Similarity* | Pregenual Anterior Cingulate | 1079 | 16 | 44 | 2 |
| | Left Supplementary Motor Area | 762 | −28 | 8 | 40 |
| | Right Superior Temporal Sulcus | 558 | 58 | −44 | −6 |
| | Left Temporoparietal Junction | 465 | −58 | −52 | 40 |
| | Right Temporoparietal Junction | 298 | 54 | −48 | 34 |
| *Latent Structure* | Right Anterior Insula/Inferior Frontal Gyrus | 696 | 34 | 16 | −10 |

Cluster size reported in voxels (2 mm$^3$). Coordinates refer to peak voxel in Montreal Neurological Institute space.

In contrast, the latent structure learning model parametric modulator identified only a cluster in right anterior insula (rAI) that extended into the inferior frontal gyrus (IFG pars orbitalis; *Figure 4*, red). This rAI cluster did not overlap with any of the clusters identified by the previous two models. Of note, this rAI cluster overlapped with a separately identified rAI cluster associated with non-social cluster assignment updating (*Tomov et al., 2018*). To provide a description of the similarity: our rAI

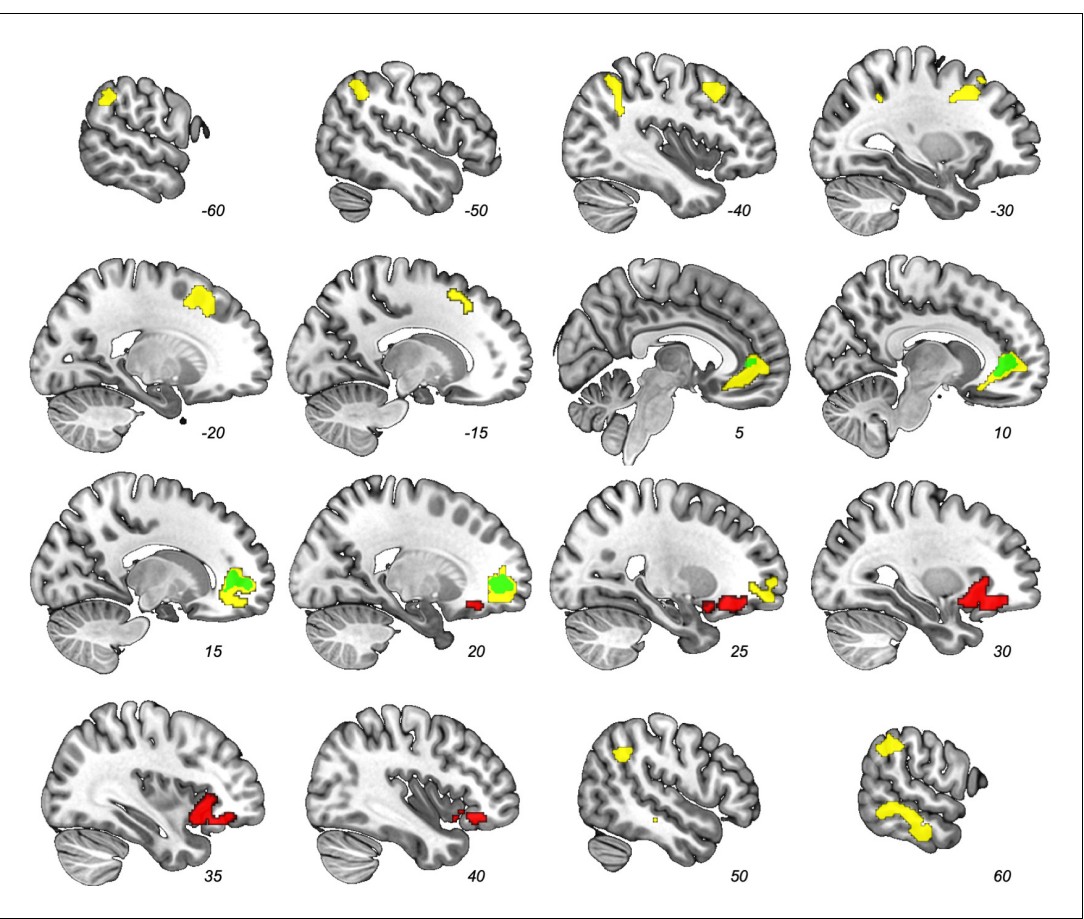

**Figure 4.** Results from whole-brain contrast (FWE-corrected p<0.05) of parametric modulators. Dyadic similarity model (green), feature similarity model (yellow), and latent structure model (red). Note the overlap between the dyadic similarity and feature similarity models in the pgACC (e.g., at x = 10).

The online version of this article includes the following figure supplement(s) for figure 4:

**Figure supplement 1.** Overlap between latent structure model parametric modulator and a separately derived ROI.

overlapped with 44.7% of that rAI result (i.e., number of common voxels across both ROIs divided by total number of rAI voxels in *Tomov et al., 2018*; *Figure 4—figure supplement 1*). Activity in this independently defined ROI correlated significantly with our latent group model, cluster-level FDR-corrected $q$ = 0.014.

## Testing the specificity of the rAI result

We conducted a k-fold cross validation (leave one out procedure) and tested whether the latent structure learning model (compared to the other models) explained more variance in the rAI. To do this, we iteratively generated rAI and pgACC clusters by conducting second-level analyses using all participants but one and tested the fit of the models in each cluster on the left-out participant. For each iteration, we computed a Bayesian information criterion for each model as that particular model score and multiplied it by −0.5 to convert it into log model evidence. We used this calculated log model evidence (one for each model for each fold) for Bayesian model selection and calculated protected exceedance probabilities (PXP) and Bayesian omnibus risk (BOR; *Rigoux et al., 2014*). A PXP reflects the probability that a particular model is more frequent in the population compared to the other models considered (beyond what would be expected by chance), while BORs reflect the probability that all model frequencies are equal to one another. To put these results into context, a previous ROC analysis found the disambiguation threshold, or the point at which we can best discriminate between $H_0$ (that both models are equally represented) and $H_1$ (that one model is represented more so than another), to exist somewhere around 50% for PXPs and around 0.25 for BORs (*Rigoux et al., 2014*). The PXP in the rAI was 82.34% for the latent structure model, but only 6.44% for the dyadic similarity model and 11.23% for the similarity-over-agents model. The BOR was 0.190. In sum, the latent structure model explained significant variance in the rAI.

On the other hand, when using the same method to test the specificity of the dyadic similarity model in the pgACC, we found that the PXPs were 51.58% for the latent structure model, 23.31% for the dyadic similarity model, and 25.11% for the similarity-over-agents model. The BOR was 0.675. In other words, for the pgACC, no single model was especially frequent over the other two.

## Predicting ally-choice behavior from neural activity

We also tested whether variability in the brain signal from our resulting ROIs would help predict variability in participants' behavior during the ally-choice trial. In other words, we asked whether the neural 'noise' from our clusters improved prediction of participant choice above and beyond mere model predictions.

We first decoded the neural signal in each ROI—pgACC and rAI—corresponding to the parametric modulator by removing the variance corresponding to other regressors in our model (see Materials and methods). We then isolated the signal corresponding to the temporal onsets of the photos of Agents A and B, which appeared right before each ally-choice trial. For each agent, we averaged the signal of interest across voxels within each ROI, respectively (i.e., we calculated the signals corresponding to Agent A and Agent B for the pgACC and the signals corresponding to Agent A and Agent B for the rAI). For each ROI, we then calculated the log difference between the signals corresponding to each agent.

We tested whether this signal would improve the fit of a logistic regression that modeled ally-choice behavior against model predictions from each of our models. Model predictions were calculated as the log difference between either the similarity with A and the similarity with B (e.g., in the case of the dyadic similarity model) at the end of the eight learning trials or the probability of the participant being grouped with A and the probability of the participant being grouped with B (in the case of the latent structure model) at the end of the eight learning trials. Log difference of the signals was orthogonalized with respect to the log difference of corresponding model predictions. Note that our parametric modulator ROIs were identified by fitting the signal during the learning phase of each run, whereas in the behavior prediction analyses here, we used signal from the ally-choice phase. As such, there is no circularity in this analysis.

While a likelihood ratio test showed that adding the signal from the rAI cluster helped to better predict variability in choice behavior for the latent structure model ($\chi^2(1)$ = 5.312, p=0.021), neither the addition of the signal from the pgACC to the dyadic similarity model ($\chi^2(1)$ = 1.526, p=0.217),

nor the addition of signal from any of the clusters associated with the similarity-over-agents model helped improve predictions of choice variability ($\chi^2(1)$s < 2.112, $p$s >0.250).

## Discussion

Here, we used a model-based analysis to compare different accounts by which people may differentiate 'us' from 'them' and found evidence for separable neural areas tracking each concurrently. While social alignment estimates based on dyadic similarity and feature similarity-over-agents recruited the pgACC, allyship estimates via latent structure learning recruited the rAI. Additionally, signal variability in the rAI cluster, but not any other cluster identified by the two other models, during the ally-choice trials helped predict ally-choice above and beyond model predictions. Furthermore, a cross-validation demonstrated that the variability explained in the rAI by our latent structure learning model was much higher than the competing models.

Several aspects of these results merit further discussion. First, this is the only evidence of which we are aware that pgACC supports incremental revisions of representations of similarity between oneself and others, both directly and across third agents. In contrast to research that relies on preexisting knowledge about specific groups or individuals, using novel agents allowed us to examine how participants' degree of alignment changed as they learned agents' preferences over time.

Second, our rAI result is consistent with previous work on updating of non-social latent structure (*Tomov et al., 2018*). Note, though, that social categorization is distinct from other forms of categorization because it requires participants to categorize themselves (*Turner et al., 1987*). Thus, our results demonstrate that rAI is capable of learning egocentrically defined latent structures.

Anterior insula is topographically well-situated to relay social information related to coalition structure. Connectivity of the anterior insula with the anterior cingulate cortex, amygdala, and ventral tegmental area (a 'salience detection' network; *Seeley et al., 2007*) and the dorsolateral prefrontal cortex (associated with cognitive control; *Chang et al., 2013*) likely affords the flexibility required to represent context-specific coalition members—someone who may be a coalition member at a debate may not be a fellow coalition member at a sports event. AI is also heavily involved in socially-relevant computations, including but not limited to self-awareness tasks such as awareness of emotions and subjective pain, and exhibits hypoactivity in persons with autism spectrum disorder on tasks such as face processing and theory of mind (*Uddin and Menon, 2009*).

The rAI region we identified included a part of the IFG (specifically, pars orbitalis). In studies of hierarchical processing in music and language, this area has been associated with sentence comprehension (*Silbert et al., 2014*) and found to exhibit sensitivity to violations of hierarchical regularities (*Cheung et al., 2018*). This area is also involved in building up sentence structure and meaning as semantic information is processed (*Jeon and Friederici, 2013*). Additionally, the IFG has been hypothesized to represent individual episodes for later recombination (*Preston and Eichenbaum, 2013*). Just as this area may be recruited to build up the structures of sentences and tonal patterns, it may also be building up inferences of social latent structures as participants learn more about other agents' preferences.

Finally, it is worth noting that while the brain tracks both of these alignment signals, variability in the signal from only one of the regions, rAI, helped improve model predictions of behavioral choices. It is possible that had participants been asked to make a different choice (e.g., identify which agent better represented a particular trait), the pgACC signal may have been more relevant. Nonetheless, this result underscores the need to further understand how social latent structures and coalitions feature in shaping people's social choices.

Accurately distinguishing 'us' from 'them' is crucial to navigating our social lives. To do so, we could rely on computing dyadic similarity with each individual agent or across agents; however, a more sophisticated approach would be to incorporate information about how agents relate to one another in order to infer latent groups in the environment through Bayesian inference. Our approach moves beyond explicit category labels and mere similarity as the sole inputs to social group representations, appealing instead to a domain-general latent structure learning mechanism, which we demonstrate predicts ally-choice. Furthermore, we provide evidence for separable neural areas tracking each of these processes; not only do we demonstrate that the rAI tracks estimates of any agent being a fellow coalition member, but we also show that the pgACC can track the fluctuations in similarity between oneself and agents (both with individual agents and over agents) in the

environment. These findings advance our understanding of the complex processes underlying social group inference and ally selection in humans and potentially other species.

## Materials and methods

### Participant selection

We first recruited participants (N = 333) under the pretense of playing a game in lab in which they would tell us about their political issue preferences and learn about others' preferences. All participants first completed this behavioral version of the task to familiarize themselves with the task and the political issues prior to scanning. Participants completed all six runs of the task as described in the scanner procedure in the following section. The main differences between the behavioral version and the scanner version were that (i) the behavioral version allowed participants to spend as much time as needed to read the prompt before proceeding, while the scanner version limited the reading time to 6 s, (ii) the behavioral version did not allow for participants to acquaint themselves with the political issues before beginning the main task, and (iii) the response buttons differed (i.e., the behavioral version involved pushing 'E' and 'I' on a keyboard rather than '1' and '2' on a button box). We could then ensure that we were only recruiting participants who could successfully make responses within the time limit. Analysis of behavioral data from this phase can be found below.

Upon completion of the behavioral task, participants were asked if they were interested in completing a similar fMRI study for pay and asked to confirm or disconfirm a series of statements relating to qualifications for participating in an fMRI study (e.g., whether they had metal in their body, being able to lie still for over an hour, etc.). Interested participants who reported no contra-indicators were invited to participate in the scanner task.

### Behavioral data analyses for sample N = 333

In the task for the participant selection phase, we fixed Agent A's and Agent B's agreement with the participant such that each agent agreed with the participant on only four of the eight trials. Additionally, Agent C always agreed with Agent B and Agent A on five trials and three trials, respectively. To create our conditions, we varied Agent C's agreement with the participant such that Agent C agreed with the participant on either seven trials (high-C) or only one trial (low-C). Given that participants had to respond within 6 s, some participants missed ally-choice trials, and only trials where data was recorded were analyzed (high-C: 906 trials, low-C: 918 trials). We used a logistic regression to model the probability of choosing Agent B's choice in the ally-choice trial as a function of condition (high-C or low-C). Including random slopes to account for subject-level effects resulted in a singular fit of the model (i.e., overfitting), so we removed them. We did not find a significant difference between our two conditions in predicting the probability for choosing Agent B, $b$ = 0.023, Wald's $z$ = 0.246, 95% CI = [−0.161, 0.207], p>0.250.

Nonetheless, we have stated before this is a small behavioral effect when previously demonstrating it across another set of experiments (*Lau et al., 2018*). Our sample size here may have been too small to detect a difference. Additionally, because participants were inexperienced and had only 6 s to respond (whereas our previous participants had an unlimited time to respond), they may have responded differently. When we limit the analysis to only the final, sixth ally-choice trial, we see a small effect of condition on the probability of choosing Agent B, $b$ = 0.454, Wald's $z$ = 2.00, 95% CI = [0.010, 0.901], p=0.046.

Finally, our two conditions, which drastically varied Agent C's level of agreement (i.e., Agent C either mostly agreed or mostly disagreed with the participant), may have been too obvious for a version of the task that was not self-paced. One participant reported as much in the debriefing. In the actual scanner version of the study, we varied the degree to which Agents A, B, and C agreed with the participant while maintaining the same agreement relationships between the agents as found in the behavioral portion to avoid this limitation.

### fMRI participants and exclusions

From our 333 participants, we recruited 61 right-handed participants (48 female, $M_{age}$ = 21.74 years, $SD$ = 4.17) in order to achieve a sample size of at least 40 participants after exclusions. This sample size was determined based on sample sizes used in other fMRI experiments (e.g., *Tomov et al.,*

*2018*; *Cheung et al., 2018*). Two participants requested to be removed from the scanner prior to the end of the study, and two participants were excluded due to a computer crash, resulting in unrecorded responses. Prior to any data analysis, we excluded five participants who fell asleep in the scanner, eight participants for excessive head movements of 4 mm or more, one participant who correctly deduced the hypothesis of the study, and one participant for missing at least one self-response per run. This left us with a sample size of 42 participants (32 female, $M_{age}$ = 22.07 years, $SD$ = 4.82). Participants provided informed consent to participate and consent to publish; all procedures complied with Harvard University's Committee on the Use of Human Subjects' guidelines (Protocol #IRB15-2048).

## Materials

To develop stimuli, we used ISideWith.com, a website that helps people determine the political party and/or candidate with which their positions best align based on yes and no responses to nationally relevant, political issues (e.g., 'Do you support the death penalty?'). The website also aggregates survey responses and makes this data publicly available (https://isidewith.com/polls). We selected issues that had accumulated at least 500,000 votes and had the greatest agreement/disagreement discrepancies, as described in Experiment 2 in *Lau et al. (2018)*. We included the 48 issues with the lowest yes-no differences as of May 2017 in the main task (see OSF for complete materials).

On each trial, we displayed the issue as text at the top of the screen. Underneath, we signified a 'yes' or 'no' response to the issue by superimposing a green check mark or a red 'X,' respectively, atop an image representing the issue. To avoid confusion, we also displayed the words, 'YES' and 'NO', underneath the corresponding images. The order of presentation of the 48 issues as well as the sides on which the agreement positions appeared on the screen were randomized for each participant.

## Face selection

For agent pictures, we selected a total of 36 photos from the Chicago Face Database (CFD; *Ma et al., 2015*) and gender-matched agents to the participant. We extracted the pool of 'White' faces (based on CFD designations) and eliminated faces based on the norming data provided by the CFD until 18 female and 18 male faces were left.

Given that the pool of faces varies for male and female faces, face selection processes varied slightly. For female faces, at least half of the respondents had to rate the face as looking 'white', and then we eliminated any face that respondents rated as unusual compared to other white females (i.e., above the midpoint of a 7-point Likert scale ranging from 1, not at all unusual, to 7, very unusual). Using the ratings of how prototypical the faces looked compared to other white females (5-point Likert scale ranging from 1, not at all prototypical, to 5, very prototypical), we then removed the faces scoring less than one standard deviation from the average of this pool to remove non-prototypical faces. We also then removed the faces that scored one standard deviation below the mean in terms of femininity (1, not at all feminine, to 7, extremely feminine). To norm for attractiveness, we eliminated anyone who was rated as one standard deviation above and below the mean of attractiveness ratings, leaving us with 38 faces. Any face two standard deviations above or below the mean age was then removed, leaving us with 37 faces, and the youngest face was then removed to generate a set of 36 faces with an average age of 27.01 years ($SD$ = 3.81).

For male faces, we followed the same initial five steps, except we eliminated any face that scored one standard deviation above the mean in femininity ratings in order to retain the more masculine-looking faces. Of the remaining 43 faces, we then eliminated anyone who was one and a half standard deviations above or below the mean age, which left us with 38 faces. For the final step, we removed the two youngest faces in the set of faces to generate a set of 36 male faces with an average age of 28.58 years ($SD$ = 5.23).

## Pre-task instructions

After being consented, participants first completed a round of instructions guiding them through a trial. They expressed their own opinion on a topic ('Should cartoons include plotlines involving duck-hunting?') by selecting 'Yes' or 'No' and then guessed and received feedback on the opinions of

Bugs and Daffy. Participants were then guided through an ally-choice trial. We told participants that for these trials, gray boxes with question marks on them would represent two different positions on a political issue. The only information participants had about the boxes was the choices of other agents—the same ones whose preferences they had just learned. We told participants to select the box they would prefer based on the other agents' choices. Participants were then introduced to the timing of the task (see below) and completed four practice trials ('Should cartoon characters conquer Mars?', 'Should cartoon characters be allowed to pilot planes?', 'Should cartoon plotlines feature day jobs?', 'Should cartoon characters be allowed to do yoga?') with these timings in place with Bugs and Daffy again. An ally-choice trial followed these four practice trials in which participants were asked to choose between Bugs' and Daffy's mystery positions. After this, the computer displayed how many responses they successfully made within the time limits, and participants were prompted to notify the experimenter. The experimenter checked that all participants made at least 11 of the 13 responses within the time limits and probed for any remaining questions about the task. Given that participants had limited time to think about the political issue at hand and because their responses were important for generating the feedback they received from the other agents, we then gave participants a list of all of the political issues they would be seeing in the scanner prior to being scanned and asked them to review all the issues.

## Task

Each run consisted of two phases: (i) eight learning trials during which participants expressed their own preferences on a political issue and learned about three others' preferences; (ii) an ally-choice trial. Each learning trial began with the participant seeing a political issue for 6 s. They then had 4.5 s to choose whether they would support that issue; a gray rectangle appeared around their selection when they had made their choice. A gray arrow pointing to their choice appeared for 3 s to confirm their choice. Each participant then learned about the preferences of other three individuals (Agents A, B, and C) via feedback.

To avoid participants weighting information from agents who looked more similar to them, we gender-matched the agents to participants' self-reported gender and only used White faces. Participants first saw a picture of one of the agents alongside his/her name and were asked to predict the preference of that agent with regards to the issue (e.g., 'Which do you think Annie chose?'); each participant had 4.5 s to guess, and a gray rectangle appeared around their selection after they made a choice. They then learned of the agent's choice when a gray arrow pointing to one of the preferences ('Yes' or 'No') appeared for 3 s (*Figure 2A*). Participants then repeated this guess and feedback process for the other two individuals. Between each screen, a fixation cross appeared for 3 s-16 s (jittered). Jitter was generated using the optseq algorithm (http://www.surfer.nmr.mgh.harvard.edu/optseq). Once this process (political issue prompt, self-choice, confirmation of self-choice, guess and feedback for A, guess and feedback for B, and guess and feedback for C) was completed for one issue, participants started a new regular trial by repeating the process for a new issue with the same three Agents. A table consisting of eight rows and four columns displayed on the right-hand side of the screen recorded the participant's and each Agent's responses on each trial. The order of the policy positions and agents was randomized; every participant saw each of the 48 policies and 18 agents only once during the experiment. The order of the agents was also randomized across runs. Agents were randomly assigned names from a preset list of names of five-letter length.

Following the eight learning trials, participants saw one 'mystery' ally-choice trial. On these trials, participants saw pictures of Agents A and B sequentially presented in a random order, each of which was followed by a fixation cross that appeared for 3 s-16 s (jittered), prior to reaching the decision screen. On the decision screen, participants saw two boxes with question marks representing two unknown positions on a political issue. Underneath the two boxes were photos of Agents A and B and gray arrows pointing to their respective choices (*Figure 2B*). We told participants that the mystery boxes contained Agent A's and Agent B's preferred positions. Participants had to indicate which one of the two unknown positions they would rather choose (e.g., 'Which would you choose? Remember, Annie and Betsy know what's inside the boxes.'). A gray rectangle appeared around the selection after they indicated their choice. Thus, participants had to align themselves with one of the two agents. The response table summarizing participants' and all agents' preferences during the block was visible during the ally-choice trial. After the ally-choice trial, participants started a new run with a new set of three agents and a new set of eight policy positions.

Participants completed six runs in total. For each run, Agents B and C agreed on five issues; Agents A and C agreed on three issues, and Agents A and B never agreed with each other. This agreement structure made it more likely that participants would cluster Agents B and C together and that Agent C's agreement with the participant would increase the selection of Agent B on the ally-choice trial. However, agreement counts between the participant and each agent varied across blocks and participants. All agent positions were entirely randomly assigned (within the constraints of the agreement/disagreement structure). Thus, across all runs and participants, some agents expressed a mixture of highly partisan left and right beliefs. In our previous behavioral studies (*Lau et al., 2018*), Experiment 2), we found that participants still exhibited the behavior predicted by the latent structure learning model despite learning about agents with less ideologically coherent profiles (i.e., agents with preference profiles constructed from a mix of left and right partisan topics).

## fMRI data acquisition and analyses

We collected data using a 32-channel head coil in a 3.0-tesla Prisma MRI scanner (Siemens) located at the University. At the beginning of each scan session, we acquired a high-resolution T-1 weighted anatomical image (T1-MPRAGE, $1 \times 1 \times 1$ mm, parallel to the anterior commissure-posterior commissure plane) for use in registering activity to each participant's anatomy and spatially normalizing data across participants. Functional images were then acquired through six echo-planar imaging (EPI) sessions each lasting 12 min. For whole brain coverage, we acquired 69 interleaved 2.0 mm slices (repetition time = 1.5 s; echo time = 30 ms; flip angle = 75 degrees; field of view = 208 mm; matrix = $104 \times 104$; in-plane acceleration (GRAPPA) = 2; multi-band acceleration factor = 3). The multi-band EPI sequence was provided by the University of Minnesota Center for Magnetic Resonance Research (*Moeller et al., 2010*; *Feinberg et al., 2010*; *Setsompop et al., 2012*; *Xu et al., 2013*).

We conducted preprocessing and statistical analyses using SPM12 (Wellcome Trust Centre for Neuroimaging, London, UK, http://www.fil.ion.ucl.ac.uk/spm). We realigned functional images to the first volume, unwarped the functional images, segmented the structural image into its respective tissue types, and normalized the gray matter of the structural to the gray matter of a standard Montreal Neurological Institute (MNI) reference brain. The mean functional images were co-registered to the structural image, and functional images were normalized to the MNI template, resliced to 2 mm $\times$ 2 mm $\times$ 2 mm voxels, and smoothed using an 8 mm FWHM Gaussian kernel.

## fMRI analyses

We modeled data with an event-related design using a general linear model. For each of the six runs, one regressor modeling the onset of self-choice (eight onsets), a second regressor modeling the onset of guesses for each of the three agents across the eight issues (24 onsets) and a parametric modulator for the second regressor, were convolved with the canonical hemodynamic response function. The parametric modulator values indexed the specific model output (the dyadic similarity, feature similarity-over-agents, or the latent structure learning model; see Computational models below) of either the similarity between the participant and the target or the prior probability of the participant belonging to the same group as the target of the guess (i.e., Agents A, B, or C). For example, when the participant was making a guess for Agent A on the seventh trial in a block, the latent structure parametric modulator took on the value of the probability that Agent A was in the same group as the participant given previous feedback. No clusters survived correction when all three models were included as parametric modulators.

We did not orthogonalize parametric modulators with respect to the second regressor (the onsets of guesses) given that we were interested in which voxels tracked our parametric modulator values rather than the mean-centered values (*Mumford et al., 2015*). In addition, we included six nuisance regressors containing the temporal and spatial derivatives for the main regressor and six run regressors. Alignment values from the latent structure learning model and the dyadic similarity model were modeled in separate models given that each represented a different hypothesis about processes in the brain. To determine which areas of the brain tracked each these models, we then entered the images resulting from contrasting the parametric modulator against baseline into a second-level analysis treating participants as a random effect. We used a voxelwise threshold of p<0.001 and

corrected for multiple comparisons using whole-brain cluster-wise family-wise error (FWE) correction from bspmview (http://www.bobspunt.com/bspmview) at the $\alpha = 0.05$ level.

## Parametric modulator correlations

Given that all three model outputs were derived from similar inputs, the correlations among the different parametric modulators were moderate to large. The average correlation between dyadic similarity and latent structure modulators was 0.8627, with values ranging from 0.5592 to 0.9704. The average correlation between the latent structure and feature similarity-over-agents parametric modulators was 0.7889 (ranging from 0.3778 to 0.9496). Finally, the average correlation between the dyadic similarity and feature similarity-over-agents parametric modulators was 0.9614 (ranging from 0.9023 to 0.9874).

To measure collinearity between the dyadic similarity, feature similarity-over-agents and latent structure modulators, we calculated the VIF ($1/(1-R^2)$, where $R^2$ is the r-squared from regressing one parametric modulator on the other). The VIF between dyadic similarity and latent structure modulators was 5.063 (generally regarded as low collinearity), while the VIF between dyadic similarity and feature similarity-over-agents parametric modulators was 13.786.

## Computational models

Dyadic similarity, $S_d$, is calculated as a function of the number of previous agreement instances between the agent and the participant divided by the number of trials elapsed, where priors for the first trial are 0.50 for each agent:

$$S_d(agent, participant) = \frac{\sum_{i=0}^{n-1} Agreement_{agent} + 1}{n+1} \tag{1}$$

Feature similarity-over-agents, $S_f$, uses output from the dyadic similarity model to construct a similarity matrix whose values are $S_d$ between the participant and each agent as well as $S_d$ between agents. Feature similarity between a participant and a particular agent is computed as the correlation between the row of the similarity matrix representing the dyadic similarity between the participant and each agent and the row of the similarity matrix representing the dyadic similarity between that particular agent and everyone else (i.e., the participant and the other two agents; *Equation 2*). To transform these correlations to interpretable probabilities, output values were rescaled to a 0 to 1 range, and a log odds transformation (i.e., $\log(S_f) - \log(1 - S_f)$) was applied.

$$S_{d_{sim}} = \begin{bmatrix} 1 & S_d(A,p) & S_d(B,p) & S_d(C,p) \\ S_d(A,p) & 1 & S_d(A,B) & S_d(A,C) \\ S_d(B,p) & S_d(B,A) & 1 & S_d(B,C) \\ S_d(C,p) & S_d(C,A) & S_d(C,B) & 1 \end{bmatrix}$$

$$S_f(agent, participant) = corr(S_{d_{sim}})_{agent, participant} \tag{2}$$

The latent structure learning model assumes that participants infer latent group assignments (a partition of agents) based on agents' choice data. The prior distribution over group assignments is a Chinese restaurant process (*Aldous, 1985*), where the probability of partition $z = [z_1, \ldots, z_M]$ given M individuals is our prior:

$$P(z|\alpha) = \frac{\alpha^K \Gamma(\alpha) \Pi_k(T_k)}{\Gamma(M + \alpha)} \tag{3}$$

where $\alpha \geq 0$ serves as the dispersion parameter (as $\alpha$ approaches infinity, each individual is assigned to a unique group), $T_k$ is the number of individuals assigned to group $k$ and $\Gamma(\cdot)$ is the gamma function. In our modeling, we used $\alpha = 2$, though the results are relatively robust to variation in this parameter. An infinite number of groups can be generated, but a 'rich get richer' dynamic favoring more popular clusters will produce more parsimonious groupings (see *Gershman and Blei, 2012*). We can derive the posterior using Bayes' rule with observed choices $C = [c_1, \ldots, c_M]$:

$$P(z|C) = \frac{P(C|z)P(z)}{\sum_{z'} P(C|z')P(z')} \quad (4)$$

The likelihood is obtained by analytically marginalizing the latent parameters under a Dirichlet-Multinomial model:

$$P(C|z) = \int_\theta P(C|\theta, z)P(\theta)d\theta = \prod_n \prod_k \frac{\Gamma(|\chi_n|\gamma)}{\Gamma(T_k + |\chi_n|\gamma)} \prod_c \frac{\Gamma(L_{kn}^c + \gamma)}{\Gamma(\gamma)} \quad (5)$$

Where $\theta$ is a set of multinomial parameters, $|\chi_n|$ is the number of options on problem $n$ and $L_{kn}^c$ is the number of individuals assigned to group $k$ who choose stance $c$ on issue $n$. The likelihood favors group assignments for which choice patterns are similar between individuals assigned to the same group.

Parametric modulator values use the probability that the agent under consideration is in the same group as the participant and are derived as the marginal posterior probability of the relevant partitions:

$$P(z_a = z_p|C) = \sum_k P(z_a = k|C)P(z_p = k|C) \quad (6)$$

## Neural signal decoding

The neural signal of interest can be algebraically derived from the standard GLM with L2-norm regularization:

$$\widehat{Signal}_{interest} = \left( Y - \sum_{i \neq interest} \hat{\beta}_i X_i \right) \frac{\hat{\beta}_{interest}}{\hat{\beta}_{interest}^2 + \lambda I} \quad (7)$$

where $Y$ is the overall signal from the voxel and $X$ is the corresponding vector from the original design matrix. For these analyses, we set our regularization parameter, $\lambda$, to a value of 1 following *Tomov et al. (2018)*.

## Acknowledgements

We thank Zachary A Ingbretsen for his assistance in programming this experiment and Momchil S Tomov for assistance in the cross-validation and analysis of neural signal predicting behavior. The research was funded by the National Institutes of Health Shared Instrumentation Grant Program (grant number S10OD020039).

## Additional information

### Funding

| Funder | Grant reference number | Author |
|---|---|---|
| Harvard University | Mind Brain and Behavior Initiative - Faculty Grant | Mina Cikara<br>Samuel J Gershman |
| National Science Foundation | BCS-1653188 | Mina Cikara |

The funders had no role in study design, data collection and interpretation, or the decision to submit the work for publication.

### Author contributions

Tatiana Lau, Conceptualization, Data curation, Formal analysis, Funding acquisition, Investigation, Visualization, Methodology, Project administration; Samuel J Gershman, Conceptualization, Formal analysis, Supervision, Funding acquisition, Investigation, Methodology; Mina Cikara, Conceptualization, Supervision, Funding acquisition, Investigation, Methodology

## Author ORCIDs
Tatiana Lau (iD) https://orcid.org/0000-0002-0681-7295
Samuel J Gershman (iD) http://orcid.org/0000-0002-6546-3298
Mina Cikara (iD) https://orcid.org/0000-0002-6612-4474

## Ethics
Human subjects: Participants provided informed consent; all procedures complied with Harvard University's Committee on the Use of Human Subjects board's guidelines. (Protocol #IRB15-2048).

## Decision letter and Author response
Decision letter https://doi.org/10.7554/eLife.53162.sa1
Author response https://doi.org/10.7554/eLife.53162.sa2

# Additional files

## Supplementary files
• Transparent reporting form

## Data availability
All materials, data, and analyses can be accessed on the Open Science Framework at https://osf.io/3wtbg/. Whole-brain maps presented in Fig. 4 can be found at https://neurovault.org/collections/6556/.

The following datasets were generated:

| Author(s) | Year | Dataset title | Dataset URL | Database and Identifier |
|---|---|---|---|---|
| Lau T, Gershman SJ, Cikara M | 2019 | Social Structure Learning in Human Anterior Insula | https://osf.io/3wtbg/ | Open Science Framework, 3wtbg |
| Lau T, Gershman SJ, Cikara M | 2020 | Social Structure Learning in Human Anterior Insula | https://neurovault.org/collections/6556/ | NeuroVault, 6556 |

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
