## [Decision Letter]

**Acceptance summary:**

Authors leveraged a combination of computational modelling and fMRI to understand how social structures are learned.

Since the time of Francis Bacon and the birth of the scientific method, researchers have recognized that the words we use to describe nature have the power to either illuminate or obfuscate, and that "many of the theoretical disputes that arise in psychology [and allied fields] are…driven…by tacit differences in terminology that ramify as substantive disagreements" (Poldrack and Yarkoni, 2016). Vague concepts and imprecise nomenclature are impediments to understanding the psychological processes that underlie human behavior, experience, and neuropsychiatric disease. Computational approaches have the potential to mitigate conceptual confusion, resolve theoretical disputes, bring the underlying neurobiology into sharper focus, and provide a common framework ('lingua franca') for integrating paradigms, species, and investigators. This approach also promises to provide specific, falsifiable predictions about the regions that collectively make up the social brain.

Here, the authors focus on how we identify a potential ally in a social situation. The authors' latent grouping structure model that assumes that we are making inferences about how agents are clustered together given the observations we have made about them. Participants made choices on political issues and predicted choices of other agents on the same issues. In subsequent ally-choice trials, they are presented with two of the three agents and decide with whom they side on a mystery choice issue. In these latter trials, the participant's choices not only depended on their previous dyadic agreements with the other agents, but also on the choices made by another agent who is not even a possible ally choice on that trial. The authors follow up on this phenomenon using fMRI and identify pregenual ACC and anterior insula clusters relating to dyadic/feature similarity and the latent grouping variable, respectively. Activity in the latter helps predict ally choices behaviorally.

**Decision letter after peer review:**

Thank you for submitting your article "Social structure learning in human anterior insula" for consideration by *eLife*. Your article has been reviewed by Dr. Frank (Senior Editor), Dr. Shackman (Reviewing Editor) and two reviewers. The following individuals involved in review of your submission have agreed to reveal their identity: Mark Thornton (Reviewer #1).

Summary:

The reviewers and I are enthusiastic about the manuscript.

This study uses methodologically sophisticated methods to address a question of broad general interest: how do "us" and "them" start to emerge from "me" and "you" in the social brain. I previously reviewed this paper at another venue, and the authors have already satisfactorily addressed most of the issues I raised in my initial review. As such, I believe this paper would make a valuable contribution to the literature without major modifications.

The behavioral phenomenon is interesting. It is one that is closely related to a previous behavioral report the authors have published. The neural analyses presented here are therefore the novel facet of the study.

The paper is generally well written and the authors provide a comprehensive reporting of the methods. Methods appear rigorous. Resource sharing via OSF is fabulous! Use of Vogt nomenclature for ACC/MCC is appreciated.

Essential revisions:

1) Testing model differences

A) Models are partly informed by the same variable: the number of previous agreements with the agent. How collinear are the model regressors in the design matrix? (A reviewer noted that the GLMs use either parametric modulators from one model or the other, and argued that inclusion of both in the same model w/o orthogonalization would provide a sharper dissociation).

B) Authors quantitatively test the model specificity of the rAI cluster. Can the authors show that the differences in exceedance probabilities between the models are meaningful?

C) Could they do the same sort of inferential testing on models differences for pgACC?

D). The modelling section is disconnected from the behavioral results. The behavioral results consist of a GLM predicting ally choices as a function of the agreement with the presented and unpresented agents. While it is interesting to find an effect of the unpresented agent, it should be more clearly described how this is explained by the computational model. A model comparison favoring the latent structure model over the other variants should be presented. Is it possible to identify trials on which the two models make different behavioral predictions? Is it possible to examine activity in the two brain areas of interest on those trials?

2) Did the authors analyze participants' behavioral responses during the learning trials? Not essential, but I think doing so could add complementary evidence for the authors' claims in at least two ways.

A) Presumably, participants are learning about the target people, and hence making more accurate guesses as they progress. (See also the more specific concerns about the task and what is being learnt, below). Even if the political issues were not consistently ideologically assigned to A or B, participants could still learn as follow: if you've seen that A and B always disagree on seven trials, you can probably guess that they will also disagree on the eight, so once you see one of their choices, you can guess the other.) If so, it would be interesting to examine systematic errors in guessing to see if they reflect the emergence of latent groups.

B) Participants guesses are revealed as either right or wrong via feedback. To the extent that they are guessing wrong, this would permit the authors to model the associated prediction errors in the brain. To the extent that the different models make different predictions, these errors would also be different, and hence lead to distinguishable brain activity.

---

## [Author Response]

Essential revisions:1) Testing model differencesA) Models are partly informed by the same variable: the number of previous agreements with the agent. How collinear are the model regressors in the design matrix? (A reviewer noted that the GLMs use either parametric modulators from one model or the other, and argued that inclusion of both in the same model w/o orthogonalization would provide a sharper dissociation).

To measure collinearity, we calculated VIF (1/(1-R^2^), where R^2^ is the r-squared from regressing one parametric modulator on the other). The VIF between the dyadic and latent structure parametric modulators was 5.063. While this does not exhibit low collinearity (generally regarded as less than 5), it also does not exhibit high collinearity (generally regarded as greater than 10). Perhaps unsurprisingly, collinearity between the dyadic and feature similarity-over-agents parametric modulators was 13.786. We have added the following to the subsection “Parametric Modulator Correlations”:

“To measure collinearity between the dyadic similarity and latent structure modulators, we calculated the VIF (1/(1-R2), where R2 is the r-squared from regressing one parametric modulator on the other). The VIF between dyadic similarity and latent structure modulators was 5.063 (generally regarded as low collinearity), while the VIF between dyadic similarity and feature similarity-over-agents parametric modulators was 13.786.”

With regards to the orthogonalization, we would argue that it is plausible that a neural signal encodes one of the variables or the other but not both; putting both into the GLM is not sensible on a priori grounds. That said, when we do a k-fold cross-validation (similar to what was done to test the specificity of the rAI result) in the rAI, we find that the protected exceedance probabilities for the GLM with only the latent structure parametric modulator and the GLM with both models’ parametric modulators are 71.91% and 28.09%, respectively. In the pgACC, we find that the protected exceedance probabilities for the GLM with only the dyadic similarity parametric modulator and the GLM with both models’ parametric modulators are 59.59% and 40.41%, respectively. Given that the single regressor model is better than the double regressor model in explaining the data, we argue that modelling the fMRI data with both parametric modulators does not make sense on empirical grounds.

B) Authors quantitatively test the model specificity of the rAI cluster. Can the authors show that the differences in exceedance probabilities between the models are meaningful?

Thank you for suggesting this. A previous ROC analysis concluded that disambiguation thresholds (the point at which the probability of confusing H0 and H1 is minimal and best discriminates between the two) for protected exceedance probabilities lies around 50 percent. This analysis additionally found that a Bayesian omnibus risk (BOR), which is akin to an error rate, of around 0.25 to be strong evidence in favor for one model over the other. We have added explanations and the BOR statistic to the section to clarify (subsection “Testing the specificity of the rAI result”).

“We used this calculated log model evidence (one for each model for each fold) for Bayesian model selection and calculated protected exceedance probabilities (PXP) and Bayesian omnibus risk (BOR; Rigoux, Stephan, Friston, Daunizeau, 2014). A PXP reflects the probability that a model is more frequent in the population compared to other models considered (beyond what would be expected by chance), while BORs reflect the probability that all model frequencies are equal to one another. To put these results into context, a previous ROC analysis found the disambiguation threshold, or the point at which we can best discriminate between H0 (that both models are equally represented) and H1 (that one model is represented more than another), to exist somewhere around 0.50 for PXPs and around 0.25 for BORs (Rigoux et al., 2014). The PXP in the rAI was 82.34% for the latent structure model, but only 6.44% for the dyadic similarity model and 11.23% for the similarity-over-agents model. The BOR was 0.190. In sum, the latent structure model explained significant variance only in the rAI.”

C) Could they do the same sort of inferential testing on models differences for pgACC?

We have added the following to subsection “Testing the specificity of the rAI result”:

“On the other hand, using the same method for testing the specificity of the dyadic similarity model in the pgACC, we found that the probability that a particular model is more frequent in the pgACC in the population than the other models considered was 51.58% for the latent structure model, 23.31% for the dyadic similarity model, and 25.11% for the similarity-over-agents model. The BOR was 0.675. In other words, no single model was especially reflected over the other two in the pgACC.”

D). The modelling section is disconnected from the behavioral results. The behavioral results consist of a GLM predicting ally choices as a function of the agreement with the presented and unpresented agents. While it is interesting to find an effect of the unpresented agent, it should be more clearly described how this is explained by the computational model. A model comparison favoring the latent structure model over the other variants should be presented. Is it possible to identify trials on which the two models make different behavioral predictions? Is it possible to examine activity in the two brain areas of interest on those trials?

The dyadic similarity and latent structure learning model make different predictions when agreement with the participant is 4 stances for Agent B and Agent A each. The number of trials in which this is true is small (n = 86) and would not give us enough power to examine the brain activity in those particular trials.

To clarify why the behavioral results are explained by the computational model, we have added the following (subsection “Computational models and neuroimaging data”):

“While a dyadic similarity model would not predict that the level of agreement with Agent C should matter for choosing on the ally-choice trial, the latent structure model does predict that Agent C’s level of agreement with the participant should matter in whether or not participants choose Agent B on the ally-choice trial. Indeed, any difference in choice behavior as a result of Agent C’s level of agreement is already inconsistent with the dyadic similarity account. Disambiguation between these two accounts of ally-choice behavior have been demonstrated previously with model simulations and behavioral studies (Gershman, Pouncy and Gweon, 2017; Lau et al., 2018).”

2) Did the authors analyze participants' behavioral responses during the learning trials? Not essential, but I think doing so could add complementary evidence for the authors' claims in at least two ways.

No, we did not; please see our responses to the next two points.

A) Presumably, participants are learning about the target people, and hence making more accurate guesses as they progress. (See also the more specific concerns about the task and what is being learnt, below). Even if the political issues were not consistently ideologically assigned to A or B, participants could still learn as follow: if you've seen that A and B always disagree on seven trials, you can probably guess that they will also disagree on the eight, so once you see one of their choices, you can guess the other.) If so, it would be interesting to examine systematic errors in guessing to see if they reflect the emergence of latent groups.

We thank the reviewers for suggesting this analysis, though we hasten to note that we would notpredict that participants would make more accurate guesses over the course of a run. Specifically, if a participant feels drawn to Agent B by virtue of Agent C’s placement, then the participant should be overestimating B’s agreement with them, if anything.

That said, we did not instruct or incentivize participants to employ a positive test strategy in their guesses in the learning portion of the task. Irrespective of how participants guessed (with or without an accuracy motive), what we know for certain is that they all received feedback about the other agents’ preferences. If participants weren’t encoding that information at all then they should have been indifferent between A and B on the ally-choice trials (i.e., selecting allies randomly), which they were not. More specifically, their responses varied in the predicted pattern as a function of Agent C’s positions.

B) Participants guesses are revealed as either right or wrong via feedback. To the extent that they are guessing wrong, this would permit the authors to model the associated prediction errors in the brain. To the extent that the different models make different predictions, these errors would also be different, and hence lead to distinguishable brain activity.

We thank the reviewers for this idea. The challenge is that the experiment wasn't designed with this analysis in mind. Assuming it takes several rounds of issues to start to accumulate the latent grouping structure within a run, we would be restricted to looking at prediction errors in the last few trials of each of six runs, and therefore underpowered to detect this result.